# Berbamine Hydrochloride Inhibits African Swine Fever Virus Infection In Vitro

**DOI:** 10.3390/molecules28010170

**Published:** 2022-12-25

**Authors:** Junhai Zhu, Lihong Huang, Fei Gao, Weijun Jian, Huahan Chen, Ming Liao, Wenbao Qi

**Affiliations:** 1College of Veterinary Medicine, South China Agricultural University, Guangzhou 510642, China; 2African Swine Fever Regional Laboratory of China (Guangzhou), Guangzhou 510642, China; 3Key Laboratory of Zoonoses, Ministry of Agriculture and Rural Affairs of the People’s Republic of China, Guangzhou 510642, China; 4National and Regional Joint Engineering Laboratory for Medicament of Zoonoses Prevention and Control, National Development and Reform Commission of the People’s Republic of China, Guangzhou 510642, China; 5Key Laboratory of Zoonoses Prevention and Control of Guangdong, Guangzhou 510642, China

**Keywords:** African swine fever virus, berbamine hydrochloride, antiviral activity, natural products

## Abstract

African swine fever virus (ASFV) causes a viral disease in swine with a mortality rate of approximately 100%, threatening the global pig industry’s economic development. However, vaccines are not yet commercially available, and other antiviral therapeutics, such as antiviral drugs, are urgently needed. In this study, berbamine hydrochloride, a natural bis-benzylisoquinoline alkaloid isolated from the traditional Chinese herb Berberis amurensis, showed significant antiviral activity against ASFV. The 50% cytotoxic concentration (CC_50_) of berbamine hydrochloride in porcine alveolar macrophages (PAMs) was 27.89 μM. The antiviral activity assay demonstrated that berbamine hydrochloride inhibits ASFV in a dose-dependent manner. In addition, a 4.14 log TCID_50_ decrease in the viral titre resulting from non-cytotoxic berbamine hydrochloride was found. Moreover, the antiviral activity of berbamine hydrochloride was maintained for 48h and took effect at multiplicities of infection (MOI) of 0.01, 0.1, and 1. The time-of-addition analysis revealed an inhibitory effect throughout the entire virus life-cycle. A subsequent viral entry assay verified that berbamine hydrochloride blocks the early stage of ASFV infection. Moreover, similar anti-ASFV activity of berbamine hydrochloride was also found in PK-15 and 3D4/21 cells. In summary, these results indicate that berbamine hydrochloride is an effective anti-ASFV natural product and may be considered a novel antiviral drug.

## 1. Introduction

African swine fever virus (ASFV), the etiological agent of African swine fever (ASF), causes a viral disease in swine with a high mortality rate approaching 100%. The ASFV was first confirmed in Kenya in 1921 and then spread to European, American, and Asian countries in the absence of any definite commercial vaccines and antiviral drugs [1,2,3]. The first ASFV outbreak in China was reported in August 2018 and resulted in a huge economic loss of several billion dollars, threatening the pig industry and economic development [4,5].

The ASFV is the only member of the family *Asfarviridae* and the only known DNA arbovirus. The ASFV is a large double-stranded DNA virus containing a linear DNA genome of 170 to 190 kb, which encodes over 150 open reading frames (ORFs) [6]. The ASFV is an enveloped virus composed of five layers: the outer membrane, capsid, inner membrane, core-shell, and nucleoid [7]. The not yet fully elucidated infection and pathogenesis mechanisms of ASFV are largely due to the complicated viral morphology, seriously hampering the progress of vaccine development [8]. With no effective commercially available vaccine, novel antiviral strategies are urgently needed.

Several methods or compounds, including nucleoside analogs, interferons, antibiotics, small interfering RNAs, CRISPR/Cas9, and plant-derived compounds, have been established with in vitro anti-ASFV activity in the past few decades [9,10,11,12,13,14,15,16,17,18,19]. Among them, natural compounds derived from plants exhibit multiple potential targets, bioactivities, and limited toxicity, which occupy more than half of the approved drugs or drug candidates in the past three decades [20,21].

Berbamine hydrochloride is a bis-benzylisoquinoline alkaloid isolated from the traditional Chinese herb Berberis amurensis. Berbaminebe was demonstrated to have antiviral activity against different types of viruses, such as bovine viral diarrhea virus (BVDV), Japanese encephalitis virus (JEV), and severe acute respiratory syndrome coronavirus 2 (SARS-CoV-2) [22,23,24,25]. Moreover, berbamine regulates multiple cellular biological processes. For example, Wang et al. showed that berbamine hydrochloride inhibits lysosomal acidification in human lung carcinoma cells [26]. Chen et al. reported berbamine’s anti-inflammatory activity and its effects on the NF-kappa B and MAPK signaling pathways [27]. Thus, we supposed that berbamine hydrochloride might be a potential broad-spectrum antiviral drug with antiviral activity against the African swine fever virus.

In this study, we investigated the antiviral activity of berbamine hydrochloride against the highly virulent genotype II ASFV strain GZ201801. We found that berbamine hydrochloride inhibited ASFV dramatically with limited cytotoxicity and showed great clinical application prospects.

## 2. Results

### 2.1. Berbamine Hydrochloride Exhibited Limited Cytotoxicity in PAMs

We explored the cell toxicity of berbamine hydrochloride (Figure 1a) in porcine alveolar macrophages (PAMs) with an MTT assay. As shown in Figure 1b, berbamine hydrochloride showed no cytotoxic effects below 5 μM. The 50% cytotoxic concentration (CC_50_) of berbamine hydrochloride was 27.89 μM (Figure 1c). The dose-dependent cytotoxicity of berbamine hydrochloride in PAMs was observed under a phase-contrast microscope. As shown in Figure 1d, cytotoxicity, including the rounding up, swelling, and detachment of cells from the culture dish, can be seen clearly at concentrations above 60 μM. These results demonstrated that berbamine hydrochloride shows limited cytotoxicity in PAMs.

### 2.2. Berbamine Hydrochloride Inhibited ASFV Infection in a Dose-Dependent Manner

A reverse transcription quantitative polymerase chain reaction (RT-qPCR) was employed to evaluate the ASFV B646L (encoding the ASFV late expression protein p72) and CP204L (encoding the ASFV early expression protein p30) transcription levels in ASFV infected cells treated with different concentrations (0 μM, 0.3 μM, 0.6 μM, 1 μM, and 5 μM) of berbamine hydrochloride. As shown in Figure 2a, the transcription levels of the ASFV B646L and CP204L genes both decrease in a dose-dependent manner. Moreover, berbamine hydrochloride markedly reduced the progeny virus titers of ASFV, and 5 μM of the berbamine hydrochloride treatment caused a 4.14 log TCID_50_ reduction in progeny virus production (Figure 2b). A Western blot assay also demonstrated that ASFV p72 and p30 expression are inhibited by berbamine hydrochloride (Figure 2c). Moreover, an indirect immunofluorescence assay (IFA) was performed to visualize the ASFV p30 protein in PAMs with different concentrations of berbamine hydrochloride treatments. As shown in Figure 2d, berbamine hydrochloride significantly reduced ASFV p30 expression in a dose-dependent manner. In summary, berbamine hydrochloride inhibited ASFV infection in a dose-dependent manner.

### 2.3. Berbamine Hydrochloride Inhibited ASFV Replication

To investigate whether berbamine hydrochloride inhibited ASFV replication, an RT-qPCR, TCID_50_ assay, and Western blot were performed after berbamine hydrochloride treatment for 24 h and 48 h. Specifically, 5 μM berbamine hydrochloride was added to PAMs after the ASFV was infected for 1.5 h. The cell debris was collected for the total RNA and protein extraction, and the cell supernatant was collected for the TCID_50_ assay at 24 h post-infection (hpi) and 48 hpi. The ASFV B646L transcription (Figure 3a), virus titre (Figure 3b), and p72 protein expression (Figure 3c) were significantly compromised from 24 h to 48 h. These results demonstrated that berbamine hydrochloride inhibited ASFV replication and reinfection in PAMs.

### 2.4. Berbamine Hydrochloride Inhibited Different ASFV Challenge Doses in PAMs

Next, we further evaluated the antiviral activity of berbamine hydrochloride against different ASFV challenge doses in PAMs. The PAMs were infected by ASFV at different multiplicities of infection (MOI) for 1.5 h, and the virus was changed with a culture medium containing 5 μM berbamine hydrochloride for another 24 h. The RT-qPCR, TCID_50_ assay, and Western blot were performed to evaluate the antiviral efficacy. As shown in Figure 4a, the mRNA level of the ASFV B646L gene showed a dose-dependent decrease at different challenge doses (1 MOI, 0.1 MOI, and 0.01 MOI) at 24 hpi in PAMs. As shown in Figure 4b, it also reduced the ASFV virus titre and p72 protein expression level in a dose-dependent manner at 24 hpi in PAMs. Our data revealed that the inhibition of berbamine hydrochloride on the ASFV was independent of the challenge dose.

### 2.5. Berbaminebe Hydrochloride Inhibited Both the ASFV Entry and Post-Entry Stages

A time-of-addition analysis was performed to investigate the inhibitory stage of berbamine hydrochloride in the ASFV life-cycle. Once the ASFV enters the cell, the progeny virus is released at 24 hpi [28]. A schematic diagram of the experimental design is shown in Figure 5a based on the ASFV life-cycle. The PAMs were incubated with 5 μM berbamine hydrochloride in the following experimental groups: the control group (PAMs were infected with ASFV for 1.5 h without berbamine hydrochloride treatment), the full-time group (PAMs were treated throughout the ASFV life-cycle (−2 to 24 hpi)), the during-time group (PAMs were treated only at the ASFV entry stage (0–1.5 hpi)), and the post-time group (PAMs were treated only at the post-entry stage (1.5–24 hpi)). The RT-qPCR demonstrated that the ASFV B646L and CP204L transcription levels decreased dramatically in all three experimental groups (Figure 5b). The progeny viruses of berbamine hydrochloride-treated or untreated were titrated with an IFA, and the virus production of all berbamine hydrochloride-treated groups was reduced significantly compared with that of the control group (Figure 5c). Accordingly, the antiviral activity of berbamine hydrochloride is affected by both the ASFV entry and post-entry stages.

### 2.6. Berbaminebe Hydrochloride Inhibited the Early Stage ASFV 

To evaluate the antiviral effect of berbamine hydrochloride in the ASFV’s early stages, we first assessed the antiviral activity of the PAMs pre-treatment with the drug against the ASFV. The PAMs were incubated with or without 5 μM berbamine hydrochloride for 4 h prior to ASFV (MOI = 0.1) infection for 24 h in the absence of the drug. As shown in Figure 6a, the transcription level of the ASFV B646L gene was inhibited in berbamine hydrochloride-treated PAMs.

Then, a viral entry assay was performed to investigate the antiviral activity of berbamine hydrochloride toward ASFV attachment and internalization. For attachment, the PAMs were infected with a mixture of the ASFV and berbamine hydrochloride at 4 °C for 1.5 h to allow virus binding but not entry. For internalization, the PAMs were infected with ASFV at 4 °C for 1.5 h and then treated with berbamine hydrochloride at 37 °C for 1 h. As shown in Figure 6b, berbamine hydrochloride inhibited both the virus attachment and internalization stages. These results indicate that berbamine hydrochloride might regulate cellular processes and inhibit early viral stages.

### 2.7. Berbamineberbamine Hydrochloride Inhibited the ASFV in PK-15 and 3D4/21 Cells

The hosts of ASFV include domestic pigs, wild boars, warthogs, bush pigs, soft ticks, and porcine monocytes; macrophages are the main target cells of the ASFV [28]. Different types of porcine cell lines, i.e., porcine kidney epithelial cell PK-15 and immortalized porcine alveolar macrophages 3D4/21, were recruited to evaluate the antiviral activity of berbamine hydrochloride against the ASFV. An MTT assay was performed to evaluate the cytotoxicity of berbamine hydrochloride in PK-15 and 3D4/21 cells. As shown in Figure 7a,b, berbamine hydrochloride exhibited similar cytotoxicity in PK-15 and 3D4/21 cells, and the CC_50_ for berbamine hydrochloride in these cell lines was 51.94 μM and 57.79 μM, respectively. To investigate the antiviral activity of berbamine hydrochloride in PK-15 and 3D4/21 cells, these cells were infected with the ASFV (MOI = 0.1) for 1.5 h, and then the virus solution was replaced by 10% FBS DMEM (for the PK-15 cells) or 10% FBS RPMI 1640 (for the 3D4/21 cells) containing 5 μM berbamine hydrochloride and incubated for 24 h. The cells were collected for the RT-qPCR. The RT-qPCR revealed that berbamine hydrochloride inhibited the ASFV B646L gene transcription level in these cells in a dose-dependent manner (Figure 7c,d). These results suggested that berbamine hydrochloride has antiviral activity on a variety of porcine cells.

## 3. Discussion

ASF is difficult to control and has caused devastating consequences over the last hundred years without any effective commercial vaccine or antiviral drug. Its complex viral structure and unexplained infection mechanisms have greatly hindered the development of vaccines and specific antiviral drugs. Several anti-ASFV small molecule compounds were synthesized and evaluated, for example, the ASFV pS273R inhibitor E-64 and microtubule-stabilizing agent compound 6b [29,30]. However, the ASFV encodes 68 structural proteins and more than 100 non-structural proteins and regulates multiple cellular biological processes to finish its life-cycle [31]. It seems difficult to achieve a strong antiviral effect against a single target.

Natural plants could be a safe and reliable source to find drugs responsible for controlling epidemic diseases. An antiviral drug derived from plants usually exhibits limited cytotoxicity and a variable antiviral mechanism. On the other hand, natural antiviral products have also shown partially broad-spectrum antiviral activity. For example, the extract of cistus incanus blocked viral binding to the cellular receptors to inhibit both human immunodeficiency virus (HIV) and filoviruses [32]. Considering that natural products have multiple targets and cellular processes’ regulatory effects, it is necessary and feasible to design anti-ASFV drugs based on natural plants or their derivatives.

Several natural products, such as toosendanin, luteolin, and apigenin, were tested for their anti-ASFV effect [24,33,34]. In this study, the authors found that berbamine hydrochloride was a novel anti-ASFV drug candidate. Berbamine hydrochloride is a bis-benzylisoquinoline alkaloid isolated from the traditional Chinese herb Berberis amurensis. Various types of viruses, including bovine viral diarrhea virus (BVDV), Japanese encephalitis virus (JEV), and SARS-CoV-2, were reported to be inhibited by berbamine [22,23,24,25]. In SARA-CoV-2, berbamine hydrochloride blocked S protein-mediated membrane fusion to execute its antiviral effects [23]. In another research article, berbamine prevented JEV entry by blocking endolysosomal trafficking [24] The ASFV encodes several proteins involved in virus entry, such as CD2v, p30, p12, pE248R, and pE199L [35,36,37,38,39]. We found that berbamine hydrochloride inhibits ASFV infection at an early stage, and berbamine hydrochloride may block these virus proteins or receptors and subsequently inhibit virus entry. Further artificial intelligence tools, such as molecular docking studies, are required to evaluate the interaction between berbamine hydrochloride and viral or host proteins. However, the receptor of ASFV is still undefined, and the structure of ASFV entry-associated proteins is not well understood.

ASFV infection regulates multiple cellular immune pathways, including native immunity, endoplasmic reticulum stress, cell apoptosis, ubiquitination, and autophagy [40]. The interactions of ASFV with host immune responses often lead to two consequences; the host recognizes the virus and deploys defense strategies to prevent viral replication and spread, or the virus escapes from the surveillance of the immune system through multiple approaches. Hence, the regulation of the host’s immune response may be considered the potential application of antiviral drugs. Berbamine was shown to pharmacologically target different cellular pathways, including the JAK-STAT, NF-κB, ERK, and AKT pathways [41,42,43]. It was reported that berbamine hydrochloride inhibits BVDV infection by regulating BVDV-induced autophagy [22]. Interestingly, autophagy inhibitors, chloroquine and wortmannin, could reduce the infections of ASFV [44,45]. In addition, NF-κB inhibitor BAY11-7082 also exhibited antiviral activity during the replication of ASFV [46]. Therefore, berbamine hydrochloride might have a potential effect on cellular immune response regulation induced by the ASFV, lead to the elimination of intracellular virus particles, and inhibit ASFV infection at the post-entry stage.

In addition to studies evaluating berbamine against disease in vitro, the effect of berbamine on the disease was also evaluated in vivo. Huang et al. found that berbamine, with a concentration of 15 mg/kg, protected mice from the lethal challenge of JEV [24]. Zhao et al. found that tumor growth was significantly inhibited when mice were treated with berbamine at a concentration of 60 mg/kg [47]. However, no study has evaluated berbamine in a pig model yet. However, cepharanthine, another bis-benzylisoquinoline alkaloid, was reported to protect piglets’ jejunum and ileum tissues from the porcine epidemic diarrhea virus (PEDV) at a concentration of 11.1 mg/kg [48]. The protective effects of berbamine hydrochloride on the ASFV should be further evaluated in pig models in the future. Nevertheless, we were unable to test the antiviral activity of berbamine hydrochloride against ASFV infection in vivo in this study.

In conclusion, our study proves for the first time that the natural product berbamine hydrochloride inhibits the ASFV with limited cytotoxicity and strong antiviral activity. Berbamine hydrochloride may be a potential antiviral drug for ASFV prevention or a cure in the future.

## 4. Materials and Methods

### 4.1. Cell Culture and Virus

Porcine alveolar macrophages (PAMs) were isolated from 4-week-old specific pathogen-free pigs and cultured in Roswell Park Memorial Institute 1640 (RPMI 1640; GIBCO, Waltham, MA, USA) medium supplemented with 10% fetal bovine serum (FBS; ExCell Bio, Shanghai, China) [49]. Porcine Kidney-15 cells (PK-15; ATCC CCL-33) and 3D4/21 cells (ATCC CRL-2843) were respectively cultured in Dulbecco’s modified Eagle’s medium (DMEM; GIBCO, Waltham, MA, USA) and RPMI 1640 medium supplemented with 10% fetal bovine serum. The ASFV strain GZ201801 (GenBank: MT496893.1) was obtained from the Research Center for African Swine Fever Prevention and Control, South China Agricultural University (Guangzhou, China).

### 4.2. Cytotoxicity Assay

Berbamine hydrochloride (Macklin, Shanghai, China; B860680) was dissolved in dimethylsulfoxide (DMSO; Sigma-Aldrich, Shanghai, China) at a concentration of 50 mM and stored at −80 °C. RPMI 1640 medium supplemented with 10% FBS was used for dilution to dissolve the berbamine hydrochloride and to use it right after it was ready. The cytotoxicity was evaluated using an MTT (3-(4,5-Dimethyl-2-Thiazolyl)-2,5-Diphenyl Tetrazolium Bromide) assay. Briefly, the PAMs and the PK-15 and 3D4/21 cells plated in 96-well plates were treated with increasing concentrations (from 1 to 100 μM) of berbamine dihydrochloride for 24 h at 37 °C in 5% CO_2_. A solution of 2% DMSO was added to the cells as the negative control. Then the culture medium was removed and replaced with 100 μL MTT (0.5 mg/mL; Sigma-Aldrich, Shanghai, China) and incubated at 37 °C for 4 h. After the removal of the supernatant, 150 μL of DMSO was added to each well to dissolve the formazan crystals for 10 min. The optical density (OD) was measured at 490 nm using a microplate reader. GraphPad Prism 8.0 (GraphPad Software, San Diego, CA, USA) was used to calculate the 50% cytotoxic concentration (CC_50_).

### 4.3. Indirect Immunofluorescence Assay (IFA)

The cells were rinsed with PBS and fixed with 4% paraformaldehyde (PFA) for 20 min, and 0.25% (*v*/*v* Triton X-100 was added to each well for 30 min at room temperature (RT) to permeabilize the cell membrane. After being rinsed with PBS three times, the cells were blocked with PBS containing 5% Bovine serum albumin (BSA) at RT for 1 h. ASFV p30 mouse monoclonal antibody diluted 1/500 (a generous gift from Daxin Peng, Yangzhou University, Yangzhou, China) was incubated at 4 °C overnight, and Alexa Fluor^®^ 488 goat anti-mouse secondary antibody diluted 1/1000 (Proteintech, Rosemont, IL, USA) was incubated at RT for 1 h. The cell nuclei were stained with 40, 6-diamidino-2-phenylindole (DAPI; Beyotime, Shanghai, China). The images were captured using a Leica DMI 4000B fluorescence microscope (Leica, Wetzlar, Germany).

### 4.4. Western Blotting Assay

The cells were lysed in RIPA lysis buffer containing 1 mM phenylmethylsulfonylfluoride (PMSF; Beyotime, Shanghai, China) at 4 °C. The supernatant was harvested by centrifugation at 10,000× *g* for 20 min at 4 °C. Equal amounts of cell lysates were loaded onto 10% sodium dodecyl sulfate-polyacrylamide (SDS-PAGE) gels and then transferred to a nitrocellulose membrane. The membranes were blocked by incubation with 5% skim milk powder for 1 h at RT. Then, the ASFV p30 or p72 mouse monoclonal antibodies (Zoonogen, Beijing, China) were incubated at 4 °C overnight and the HRP-labeled goat anti-mouse IgG (Beyotime, Shanghai, China) reagent was incubated for 1 h at RT. The results were analyzed using a Tanon- 5200 multi-infrared imaging system (Shanghai Tianneng Technology Co., Ltd., Shanghai, China).

### 4.5. Reverse Transcription Quantitative Polymerase Chain Reaction (RT-qPCR)

The RNA was extracted from the samples using a total RNA rapid extraction kit (Fastagen, Shanghai, China) and reverse transcribed as complementary DNA (cDNA) using a PrimeScript RT reagent kit with the gDNA Eraser (TaKaRa Bio, Kyoto, Japan). The gene expression in the cDNA samples was measured with the hamQ Universal SYBR qPCR Master Mix according to the manufacturer’s instructions (Vazyme Biotech, Nanjing, China). The sequences of primers used for the RT-qPCR are listed in Table 1. The RT-qPCR was performed on the CFX96 real-time PCR detection system (Bio-Rad, Shanghai, China). The RNA relative expression of each target gene was normalized to the GAPDH expression and then calculated using the 2^−ΔΔCT^ method.

### 4.6. TCID_50_ Assay

The PAMs were seeded in 96-well plates and incubated at 37 °C for 4 h. The supernatant was discarded, and the cells were infected with ten-fold serially diluted samples (test group) or RPMI 1640 medium (blank control group) for 1.5 h. Then, the samples were removed and washed with PBS. RPMI 1640 containing 10% FBS was added in each well, and the cells were cultured at 37 °C for 24 h. The test and blank control groups were set up with eight replicate wells in each group. The viruses were detected with the IFA method, and the TCID_50_ was calculated using the Reed and Muench method.

### 4.7. Antiviral Activity Assay

To investigate the antiviral activity of berbamine hydrochloride against the ASFV, the PAMs were pre-treated with berbamine hydrochloride at different concentrations (0–5 μM) for 2 h, the compound was removed, and the cells were washed three times with PBS. The ASFV at a multiplicity of infection (MOI) of 1 was then added and infected for 1.5 h. After that, the solution was discarded and washed three times. Then cells were cultured with the corresponding concentration of berbamine hydrochloride for another 72 h. The cells were collected for the RT-qPCR and Western blot analysis, and the supernatants were collected for the IFA and TCID_50_ assay.

### 4.8. Time-of-Addition Assay

The time-of-addition assay was composed of four experiments. (1) Control: PAMs were infected with the ASFV (MOI = 0.1) for 1.5 h, and then the virus solution was replaced with 10% FBS RPMI 1640 and incubated for 24 h. (2) Full-time: PAMs were pre-treated with 5 μM berbamine hydrochloride for 2 h, then the cells were infected with the ASFV (MOI = 0.1) for 1.5 h in the presence of berbamine hydrochloride at a concentration of 5 μM, then the virus solution was replaced with 10% FBS RPMI 1640 containing 5 μM berbamine hydrochloride and incubated for 24 h; (3) During-time: PAMs were infected with the ASFV (MOI = 0.1) for 1.5 h in the presence of berbamine hydrochloride at a concentration of 5 μM, then the virus solution was replaced with 10% FBS RPMI 1640 and incubated for 24 h; (4) Post-time: PAMs were infected with the ASFV (MOI = 0.1) for 1.5 h, then the virus solution was replaced with 10% FBS RPMI 1640 containing 5 μM berbamine hydrochloride and incubated for 24 h. The cells and supernatants were collected for the RT-qPCR and TCID_50_ analysis, respectively.

### 4.9. Viral Entry Assay

To investigate the inhibition of berbamine hydrochloride during ASFV attachment, PAMs were infected with the ASFV (MOI = 0.1) for 1.5 h at 4 °C in the presence of berbamine hydrochloride at a concentration of 5 μM. The virus was removed and then cells were washed with ice-cold PBS three times.

To investigate the inhibition of berbamine hydrochloride during ASFV internalization, PAMs were initially infected with the ASFV (MOI = 0.1) for 1.5 h at 4 °C to allow virus attachment but not internalization. After binding, the virus was removed, and then the cells were washed with ice-cold PBS three times. The cells were transferred to 37 °C for another 1h in the presence of berbamine hydrochloride at a concentration of 5 μM. Then, berbamine hydrochloride was discarded, and the cells were incubated for 24 h with 10% FBS RPMI 1640 at 37 °C. The cells were collected for the RT-qPCR analysis.

## 5. Statistical Analysis

All data were analyzed using GraphPad Prism 8.0 software (GraphPad Software, San Diego, CA, USA) and are presented as the mean ± standard error of the mean of at least three independent experiments. The statistical comparisons between the groups were performed using paired or non-paired *t*-tests. Two-tailed p values were determined, and a *p*-value of <0.05 was considered to indicate statistical significance (*, *p* < 0.05; **, *p* < 0.01; ***, *p* < 0.001; ****, *p* < 0.0001).

## Figures and Tables

**Figure 1 molecules-28-00170-f001:**
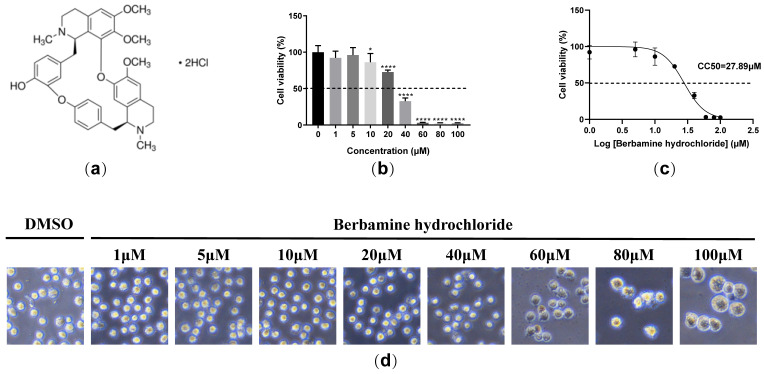
The cytotoxicity of berbamine hydrochloride in PAMs. (**a**) The chemical structure of berbamine hydrochloride; (**b**) the cell viability of berbamine hydrochloride in PAMs was evaluated with an MTT assay; (**c**) the CC_50_ of berbamine hydrochloride in PAMs was calculated with GraphPad Prism 8.0; (**d**) the effect of berbamine hydrochloride on the cell morphology of PAMs under the microscope. Each datum represents the results of three independent experiments (means ± SD). Significant differences compared with the control group are denoted with *, *p* < 0.05; ****, *p* < 0.0001 and ns denotes no significant difference.

**Figure 2 molecules-28-00170-f002:**
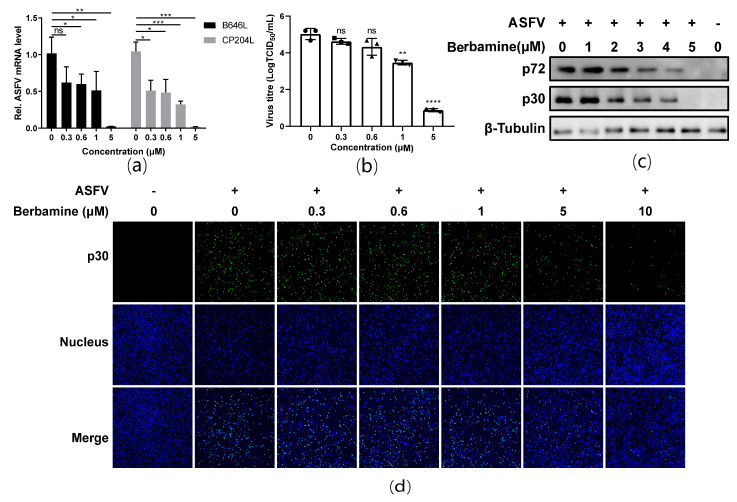
Antiviral activity of berbamine hydrochloride against the ASFV in PAMs. (**a**) ASFV B646L and CP204L gene transcription levels were evaluated with an RT-qPCR; (**b**) ASFV production was evaluated with a TCID_50_ assay; (**c**) ASFV p72 and p30 protein expression levels were evaluated with a Western blot; (**d**) ASFV p30 protein expression was evaluated with an IFA. Each datum represents the results of three independent experiments (means ± SD). Significant differences compared with the control group are denoted with *, *p* < 0.05; **, *p* < 0.01; ***, *p* < 0.001; ****, *p* < 0.0001 and ns denotes no significant difference.

**Figure 3 molecules-28-00170-f003:**
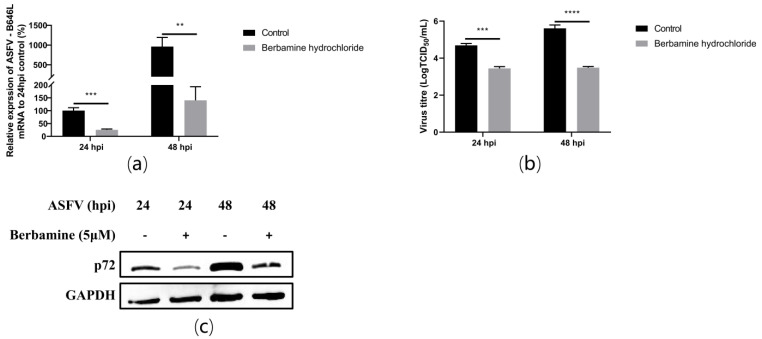
The antiviral activity of berbamine hydrochloride against ASFV replication in PAMs. (**a**) The ASFV B646L transcription level was evaluated with an RT-qPCR; (**b**) the ASFV production was evaluated with a TCID_50_ assay; (**c**) the ASFV p72 protein expression level was evaluated with Western blotting. Each datum represents the results of three independent experiments (means ± SD). Significant differences compared with the control group are denoted with **, *p* < 0.01; ***, *p* < 0.001; ****, *p* < 0.0001.

**Figure 4 molecules-28-00170-f004:**
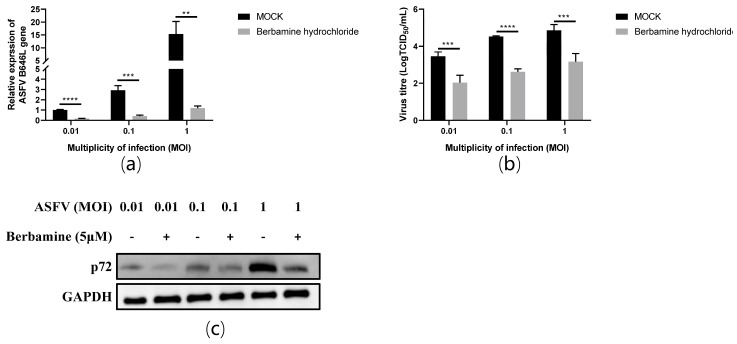
The antiviral activity of berbamine hydrochloride against the ASFV at different MOI (MOI = 0.01, 0.1, and 1) in PAMs. (**a**) The ASFV B646L transcription level was evaluated with an RT-qPCR; (**b**) the ASFV production was evaluated with a TCID_50_ assay; (**c**) the ASFV p72 protein expression level was evaluated with Western blotting. Each datum represents the results of three independent experiments (means ± SD). Significant differences compared with the control group are denoted with **, *p* < 0.01; ***, *p* < 0.001; ****, *p* < 0.0001.

**Figure 5 molecules-28-00170-f005:**
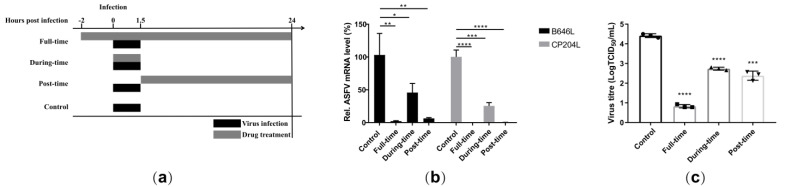
Time-of-addition analysis. (**a**) The schematic diagram of the time-of-addition; (**b**) the ASFV B646L and CP204L gene transcription levels in the different experimental groups was evaluated with an RT-qPCR; (**c**) the ASFV production in the different experimental group was evaluated with a TCID_50_ assay. Each datum represents the results of three independent experiments (means ± SD). Significant differences compared with the control group are denoted with *, *p* < 0.05; **, *p* < 0.01; ***, *p* < 0.001; ****, *p* < 0.0001.

**Figure 6 molecules-28-00170-f006:**
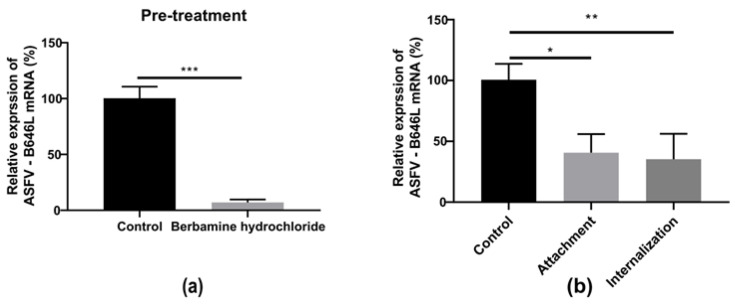
The effect of berbamine hydrochloride on the ASFV’s early stage. (**a**) The PAMs were pre-treated with berbamine hydrochloride for 4 h and then infected with the ASFV for 24 h; the ASFV B646L transcription level was evaluated with an RT-qPCR. (**b**) Attachment: the PAMs were infected with a mixture of the ASFV and berbamine hydrochloride at 4 °C for 1.5 h and incubated at 37 °C for another 24 h in the absence of drugs; internalization: PAMs were infected with the ASFV at 4 °C for 1.5 h and then treated with berbamine hydrochloride at 37 °C for 1 h, then the drug was discarded, and the solution was incubated at 37 °C for another 24 h in the absence of the drug. The ASFV B646L transcription level was evaluated with an RT-qPCR. Each datum represents the results of three independent experiments (means ± SD). Significant differences compared with the control group are denoted with *, *p* < 0.05; **, *p* < 0.01; ***, *p* < 0.001.

**Figure 7 molecules-28-00170-f007:**
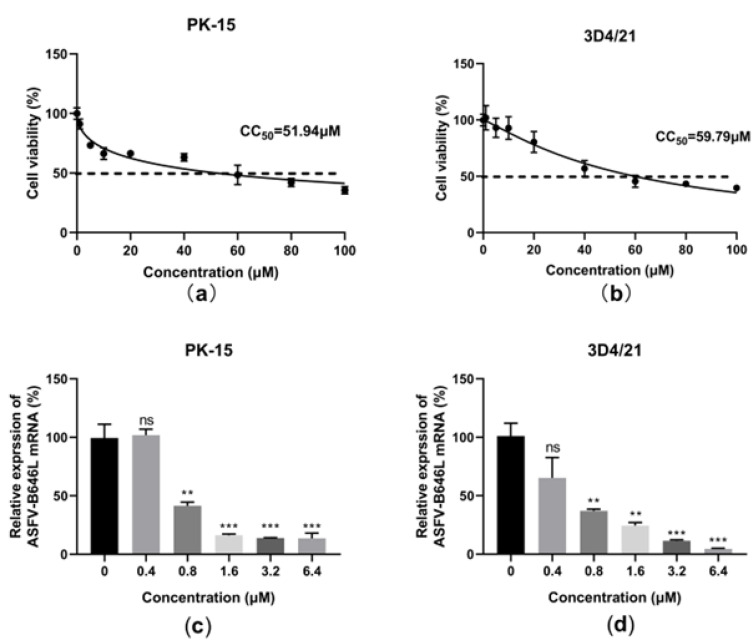
The effect of berbamine hydrochloride against the ASFV in PK-15 and 3D4/21 cells. The cell viability of berbamine hydrochloride in PK-15 (**a**) and 3D4/21(**b**) was evaluated by MTT assay and CC_50_ was calculated by GraphPad Prism 8.0. Inhibition of berbamine hydrochloride against ASFV B646L and CP204L gene transcription in PK-15 (**c**) and 3D4/21(**d**) was evaluated by RT-qPCR. Each datum represents the results of three independent experiments (means ± SD). Significant differences compared with the control group are denoted with **, *p* < 0.01; ***, *p* < 0.001 and ns denotes no significant difference.

**Table 1 molecules-28-00170-t001:** Sequences of primers used for RT-qPCR.

Gene	Sense Primer (5′-3′)	Antisense Primer (5′-3′)	Reference
B646L	CCCAGGRGATAAAATGACTG	CACTRGTTCCCTCCACCGATA	OIE
CP204L	GAGGAGACGGAATCCTCAGC	GCAAGCATATACAGCTTGGAGT	[50]
GAPDH	GAAGGTCGGAGTGAACGGATTT	TGGGTGGAATCATACTGGAACA	[51]

## Data Availability

Not applicable.

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
