# Peer review of "Berbamine Hydrochloride Inhibits African Swine Fever Virus Infection In Vitro"

_molecules, 2022, doi:10.3390/molecules28010170_

Round 1

Reviewer 1 Report

In the manuscript, authors have investigated the use of berbamine hydrochloride as an antiviral agent against ASFV infection in vitro settings. The manuscript is well written and provides sufficient data to conclude the importance of berbamine hydrochloride as an antiviral agent ASFV. I do not have any major concerns with this manuscript. It would be great to see the potency of this alkaloid in vivo settings.

Add dilution of antibodies used for IFA staining. 

Author Response

We feel great thanks for your professional review work on our article. According to your nice suggestions, we have made extensive corrections to our previous draft, the detailed corrections are listed below

Reviewer 2 Report

The manuscript entitled “Berbamine hydrochloride inhibits ASFV infection in vitro” describes an antiviral activity of Berbamine hydrochloride extracts on the ASFV genotype II currently circulating in Europe and Asia. Since African swine fever poses a particular concern for worldwide pig production sector, the subject is of high importance. The method is based on the in vitro testing of diluted extract on the virus infectivity. The study is highly novel, however the quality of manuscript preparation and result presentation must be improved. Sadly, I also have some doubts about authors’ basic virological knowledge, since it seems that they do not differentiate between cytopathic effect (CPE) and cytotoxicity, which is a key knowledge for the performed study, similar issue was for RT-qPCR, real-time PCR and quantitative PCR. The issue of the application of 3D4/21 and PK-15 established cell lines is also arguable, since they are not permissive for virulent ASFV genotype II strains isolated from the field, especially without prior adaptation, but any detailed information about the virus used for the study is missing. The discussion in my opinion is a bit too short, some issues should be developed, which is further specified. English is sufficient, but some minor errors must be improved. The study, after major improvements, will be valuable and its overall merit would be correct, but only when highlighted as very preliminary results.

Line 16: ASFV is not considered highly contagious anymore, the R0 for the disease is quite low

Line 18: They are developed, but still require conformation of their safety therefore they are not commercially available yet

Line 21: the word “dramatic” does not fit to the context, please use another word

Line 24: Log of what? TCID50 or HAD 50?

Line 25-26: I do not understand the following phrase: “showed a challenge dose-independent feature”

Line 28:” …early stage of ASFV” … replication?

Line 28-29: If so, then specify in the Abstract which ASFV strain/genotype has been used for the study, since not all of them are able to replicate in mentioned established cell lines.

Line 29-30: It is “potentially” or “may be considered” a new drug, please rephrase.

Line 36: please, check comment to line 16

Lines 43-45: The phrase is too complicated, some verbs are missing, please rephrase.

Line 238: please cite relevant work for PAM isolation

Line 240: Add PK-15 ATCC reference number

Line 242: what is 1640 medium, did you mean RPMI? Add the info about the producer to it, and to FBS.

Line 247-249: What was the dissolvent to prepare the used concentrations the commercially purchased compound? What solvent was used as a negative control?

Line 250-251: MTT test producer?

Line 254: please unify the info about producers in the whole manuscript according to the info about Graphpad, include also the info about country in the end

Line 274: qRT-PCR does NOT equal to quantitative real-time PCR! The thing that was performed is quantitative reverse transcription PCR. It must be improved thought the whole manuscript.

Line 276: reverse transcribed, not transcripted

Line 280: or quantitative, or real-time, because both means the same!

Line 287: Was the compound removed and cells washed after two hours, just before virus infection? Was the virus removed after infection (further you mention 1.5 h of infection?)

Line 289-290: Only one cell was collected? Or you have analysed the supernatant collected from the whole infected well?

Line 294: RPMI 1640 I suppose, improve it throughout the whole text

Line 314-315: cells were collected

Line 66-67: specify the genotype of the virus and its virulence, pathogenicity, it is a key information in this manuscript!

Line 70-78: It seems that You do not know the difference between cytopathic effect and cytotoxicity, which is a basic knowledge in virology, especially for the study that you performed!

Figure 1: info about statistical significance and p value is missing

Line 87-88: what concentrations have been used?

Figure 1b: how did you calculated TCID50 for PAM cells? Moreover, the information about virus titration method is missing in material and methods section

Line 91: log of what?

Line 95-96: reduced what of ASFV? Genome copies? Protein expression? Replication level? This must be clearly stated!

Figure 3a, 4a: units on the y axis?

Figure 5a, 5b – In experimental groups, the relative mRNA level does not correspond to the results from virus titration, what is the explanation for this phenomenon?

Line 176: Why these specific cell lines were selected? Please, cite at least two independent sources, which inarguably demonstrate that ASFV genotype II (but I only suppose that this variant has been used, if not, please correct me), isolated from the field, is able to replicate in PK-15 and 3D4/21 cells without previous adaptation. Please check the following recent paper: https://www.mdpi.com/1999-4915/14/12/2642. The transcription level of B646L in this cells is impaired, therefore Your results stand in the opposite to the current knowledge. Please, try to explain this phenomenon, or select the other model to confirm You thesis. Moreover, the detailed description on the methodology is lacking, please add this information in the manuscript.

Line 192: CP204L has been also used as a target? It has not been mentioned elsewhere in relation to the established cell lines study.

Line 215: Latin is not desirable in this context, please improve

Line 222: grammar “blocking”?

Line 225-226: how about post-entry stage, is there any hypothesis how berbamine blocks ASF replication? Please, develop this issue in the discussion.

The whole discussion should be slightly developed, please relate to the sources showing that berbamine works well for other disease curation, relate to any other disease and its in vivo studies, relate to the concentrations which have been tested in vivo, try to find any study that evaluated berberine or its derivatives on the pig model etc.

Lines 194-196: may be omitted since these information have been included in the introduction, but are not necessary in the discussion

Author Response

We feel great thanks for your professional review work on our article. As you are concerned, there are several problems that need to be addressed. According to your nice suggestions, we have made extensive corrections to our previous draft, the detailed corrections are listed below.

If there are any other modifications we could make, we would like very much to modify them and we really appreciate your help. Thank you very much for your help.

Round 2

Reviewer 2 Report

Thank You for taking my suggestions into account. The paper has been significantly improved. I do not have any additional comments.